# Transcription Factor Networks in Leaves of *Cichorium endivia:* New Insights into the Relationship between Photosynthesis and Leaf Development

**DOI:** 10.3390/plants8120531

**Published:** 2019-11-21

**Authors:** Giulio Testone, Elena Baldoni, Maria Adelaide Iannelli, Chiara Nicolodi, Elisabetta Di Giacomo, Fabrizio Pietrini, Giovanni Mele, Donato Giannino, Giovanna Frugis

**Affiliations:** 1Istituto di Biologia e Biotecnologia Agraria (IBBA), Operative Unit of Rome, Consiglio Nazionale delle Ricerche (CNR), Via Salaria Km. 29,300, 00015 Monterotondo Scalo (Roma), Italy; giulio.testone@cnr.it (G.T.); elena.baldoni@cnr.it (E.B.); mariaadelaide.iannelli@cnr.it (M.A.I.); chiara.nicolodi@cnr.it (C.N.); edg75@tiscali.it (E.D.G.); giovanni.mele@cnr.it (G.M.); donato.giannino@cnr.it (D.G.); 2Istituto di Biologia e Biotecnologia Agraria (IBBA), Consiglio Nazionale delle Ricerche (CNR), Via Bassini 15, 20133 Milano, Italy; 3Istituto di Ricerca sugli Ecosistemi Terrestri (IRET), Operative Unit of Rome, Consiglio Nazionale delle Ricerche (CNR), Via Salaria Km 29,300, 00015 Monterotondo Scalo (Roma), Italy; fabrizio.pietrini@cnr.it

**Keywords:** transcription factors, photosynthesis, leaf shape, plant development, *Cichorium endivia*, RNA-seq, cluster analysis, gene co-expression networks

## Abstract

*Cichorium endivia* is a leafy crop closely related to *Lactuca sativa* that comprises two major botanical varieties characterized by a high degree of intraspecific morphological variation: var. *latifolium* with broad leaves (escarole) and var. *crispum* with narrow crisp curly leaves (endive). To investigate the relationship between leaf morphology and photosynthetic activity, escaroles and endives were used as a crop model due to the striking morphological diversity of their leaves. We constructed a leaf database for transcription factors (TFs) and photosynthesis-related genes from a refined *C. endivia* transcriptome and used RNA-seq transcriptomic data from leaves of four commercial endive and escarole cultivars to explore transcription factor regulatory networks. Cluster and gene co-expression network (GCN) analyses identified two main anticorrelated modules that control photosynthesis. Analysis of the GCN network topological properties identified known and novel hub genes controlling photosynthesis, and candidate developmental genes at the boundaries between shape and function. Differential expression analysis between broad and curly leaves suggested three novel TFs putatively involved in leaf shape diversity. Physiological analysis of the photosynthesis properties and gene expression studies on broad and curly leaves provided new insights into the relationship between leaf shape and function.

## 1. Introduction

Plant growth and development are critically controlled by light. Photosynthesis is the most important light-dependent process, involving the transduction of photons absorbed by photosynthetic pigments into chemical free energy. Due to the close relationship between net photosynthesis and crop yield, enhancing photosynthetic efficiency constitutes a major target to improve crop production in a global changing climate [1].

Leaf morphological differences at inter- and intraspecific level can explain substantial variations in photosynthetic properties under different environmental constraints [2,3]. Thus, the inseparable relationship between leaf architecture and photosynthesis holds great potential to enhance photosynthetic efficiency by modifying leaf anatomy and morphology [1].

Photosynthesis is the consequence of a multistep and complex process that involves several biological pathways, and in particular: (i) the photosynthetic electron transport system (PET), in which the light energy is converted into ATP and NADPH; (ii) the Calvin–Benson cycle, a photosynthetic carbon fixation cycle in which CO_2_ is fixed into carbohydrates [4,5]. The photosynthetic process is mediated by the coordinated action of ca. 3000 different proteins, commonly referred to as photosynthesis proteins, that are encoded from both chloroplast and nuclear genomes [6,7]. Coordinating the transcription of photosynthesis-associated nuclear genes (PhANGs) and photosynthesis-associated plastid-encoded genes (PhAPGs) to maintain the right balance among nuclear encoded proteins, plastid proteins, carotenoids, and chlorophylls is critical for the correct assembly of functional photoprotective and photosynthetic complexes within chloroplasts under both optimal and suboptimal conditions [8]. A central role in the activation of photosynthesis-related genes is played by the red and far-red light-sensing phytochrome (PHY) family members [9,10]. PHYs regulate many aspects of plant growth and development in response to red and far-red light signals from the environment [11]. They are encoded in higher plants by small gene families, for example, *Arabidopsis thaliana* contains a *PHY* gene family with five members, with designations from *PHYA* to *PHYE*. Phytochromes induce major changes in the expression of genes related to different pathways, including the promotion of the photosynthetic apparatus assembly, and the production of the photosynthetic pigments, chlorophylls, and carotenoids [12,13,14]. In *Arabidopsis thaliana*, many transcription factors (TFs) have been identified to be downstream of PHY signaling [8]. Among them, the best characterized are the PHYTOCHROME INTERACTING FACTOR (PIF) and PIFLIKE (PIL) families of basic helix-loop-helix (bHLH) TFs that act mainly as negative regulators of photosynthesis gene expression in response to light availability [8]. PHYs promote the stability of TFs that regulate photosynthesis gene expression, such as *ELONGATED HYPOCOTYL 5* (*HY5*) and *HY5-HOMOLOGOUS* (*HYH*) that promote photosynthetic machinery assembly, photopigment production, chloroplast development, and seedling cotyledon expansion [15]. For a broad range of responses, PIFs act antagonistically with HY5 and repress carotenoid accumulation by down-regulating the expression of the *PHYTOENE SYNTHASE* (*PSY*) gene, the main rate-determining enzyme in the carotenoid pathway [16]. PHY-mediated PIF degradation is a central mechanism for inducing chloroplast biogenesis through the activation of specific PhANGs [17] such as *GOLDEN2-LIKE* (*GLK*), *SIGMA FACTORS* (*SIG*), and *GROWTH REGULATING FACTORS* (*GRF*) [18]. GLKs positively regulate chloroplast development and the expression of photosynthesis genes like those encoding light-harvesting complex (LHC) proteins and key enzymes of the chlorophyll biosynthetic pathway. The nuclear encoded SIGs activate the plastid-encoded RNA polymerase (PEP) that drives the expression of plastid-encoded genes for photosynthesis proteins, including core components of photosystem I (PSI) and photosystem II (PSII) [8]. GRFs regulate chloroplast division and represent an important link between photosynthesis and leaf growth. In fact, GRFs and their interacting factors (GIFs) control growth and development of leaves and cotyledons by regulating meristem function through the control of cell proliferation [19]. Plastid signals repress the induction of light-induced genes through plastid-to-nucleus retrograde signals emitted by dysfunctional chloroplasts (retrograde signaling, RS) [18,20,21]. Phytochrome and RS pathways converge antagonistically to regulate the expression of *GLK1*, i.e., normal light lead to the induction of the *GLK1* regulator whereas light at excessive levels is sensed by the plastid leading to *GLK1* repression [22].

Although the function of many photosynthetic genes is known, the dissection of the genetic networks underlying photosynthesis gene expression is challenging as they are interconnected with those that control other plant developmental and biochemical processes. Despite this complexity, about 70 transcriptional regulators which alter the expression of groups of photosynthesis genes or the photosynthetic activity in Arabidopsis have been identified [8]. The small size of this regulatory network indicates that several key players are yet to be known. Unraveling this complex regulatory network is, thus, crucial for the development of strategies for increasing photosynthetic efficiency, as well as identifying the role and importance of leaf shape in contributing to the photosynthetic output of leaves, which is largely unexplored. Whereas some studies on the relationship between leaf anatomy (i.e., chloroplast number, mesophyll cell size, and morphology) and leaf photosynthetic capacity are available [1], those dealing with the relationship between photosynthesis and leaf shape are quite limited. One study was carried out on Pelargoniums and highlighted a correlation between the degree of leaf dissection and the photosynthetic rates under a high temperature [23]. A recent work used both idealized and scanned leaves from Arabidopsis and *Cardamine hirsuta* to test whether the phyllotaxis Fibonacci golden angle was optimal for light capture [24], but no relationship with photosynthesis efficiency was analyzed. 

To investigate the relationship between leaf morphology and photosynthetic activity, we studied four *Cichorium endivia* cultivars belonging to the two major botanical varieties characterized by a high degree of intraspecific leaf morphological variation, var. *latifolium* with broad leaves (escarole) and var. *crispum* with narrow crisp curly leaves (endive). We used a *C. endivia* transcriptome and RNA-seq data from leaves of escaroles and endives to obtain a bona fide leaf database of transcription factors and photosynthesis-related genes. Cluster analysis and construction of gene co-expression networks (GCN) allowed the identification of a main photosynthesis-driven TF regulatory network with a central role in the leaf transcriptome, as well as candidate genetic hubs controlling the overall leaf transcriptional network. Morphological and physiological analysis of photosynthetic parameters were carried out to establish a possible link between photosynthesis and leaf shape. Differential expression analysis between broad- and curly-leaved plants, and a “guilt by association” approach within gene co-expression networks identified developmental genes that may play a role at the boundary between photosynthesis and leaf shape.

## 2. Results

### 2.1. Construction of TF-PHOTO, a C. Endivia Leaf Database for Transcription Factors and Photosynthesis-Related Genes 

We used the transcriptome of *C. endivia* (cv ‘Domari’), previously obtained in our lab by de novo assembly [25], and leaf RNA-seq data from two production cycles, to construct a bona fide leaf database of transcription factors and photosynthesis-related genes (TF-PHOTO) (Appendix A). First, a previous raw TF database [25], based on the TF prediction server of the PlantTFDB [26], was refined to obtain a reliable novel database including only non-redundant full-length and near full-length TF transcripts. To achieve this, the database was implemented by integrating homologues of *Lactuca sativa* TFs recently annotated in the PlantTFDB platform or published [27]. All the contigs (ca 6000) were then: (i) aligned to the released *L. sativa* genome [28] to obtain 0 to 1 hint/locus (Appendix A), (ii) filtered for full-length percentage prediction (FL%) ≥ 0.7 in protein-protein alignment with lettuce peptide database, (iii) validated as TFs with the MapMan package [29], and (iv) filtered for leaf expression in more than 90% of the RNA-seq leaf samples. Genes homologous to those involved in the transcriptional control of photosynthesis [8] were also identified in *C. endivia* and integrated in the database if not present. A total of 1560 bona fide unigenes were finally obtained, which were assigned to families according to the PlantTFDB rules [26] (Figure 1).

As for photosynthesis and photosynthesis-related genes (energy dissipation and reactive oxygen species detoxification) homologues of *Arabidopsis thaliana* genes belonging to GO:0015979 (photosynthesis), GO:0009813 (flavonoid metabolic processes), GO:0016116 (carotenoid metabolic process), and GO:0019852 (ascorbic acid metabolic process) terms were identified. These contigs were also aligned to *L. sativa* genome, filtered for expression in leaves, and added to the final TF-PHOTO database that contained a total of 1833 genes (Appendix A). 

### 2.2. Finding Expression Patterns in Leaves Using K-Means Cluster Analysis

To characterize the regulatory transcriptional networks occurring in mature leaves, we analyzed RNA-seq data from leaves of field-grown plants of four commercial cultivars of *C. endivia*, the curled-leaved endives ‘Domari’ and ‘Myrna’ and the broad-leaved escaroles ‘Confiance’ and ‘Flester’, obtained from two independent years of production, 2011 (previously unpublished) and 2012 [25]. To identify groups of genes with similar expression profiles within our TF-PHOTO database, we performed K-means clustering [30], which uses Euclidean or correlation distance as similarity measures [31]. As K-means clustering requires a priori knowledge of the number of clusters, we tested several methods and empirically established the optimum number at eight using an a posteriori validation method (Appendix A). The results of the K-means analysis are shown in Figure 2a–h. Amongst the eight clusters, cluster three (Figure 2c) contained most of the genes directly related to photosynthesis (Figure 2i, Appendix A) and was highly anticorrelated with cluster six (r = −0.97) (Figure 2f, Appendix A), which likely contained negative regulators of photosynthesis or antagonistic pathways. Cluster three and six represented the two most abundant and robust clusters (465 and 281 genes, medium correlation index with centroid = 0.87 and 0.86, respectively), which reflects the major function of leaves as photosynthetic organs. The gene expression profiles in cluster three and six pointed to a different behavior of ‘Flester’ with respect to the other three cultivars. ‘Domari’, ‘Myrna’, and ‘Confiance’ showed a strong difference in gene expression levels between the two years of production (2011 and 2012), whereas ‘Flester’ displayed similar expression levels in the two years (Figure 2c,f). Photosynthesis-related genes were generally expressed at lower levels in ‘Flester’ as compared with the other cultivars, particularly in 2011 samples. As for the other clusters, cluster five and seven (Figure 2e,g) were the least abundant and most heterogeneous, with several genes showing low correlation with the cluster core (black thick line in Figure 2). Instead, cluster one, two, and eight represented several genes that fit well with cluster assignment, with most genes showing high correlation indexes with the cluster core (Appendix A). Cluster one and eight (Figure 2a,h) were mildly correlated to cluster three, whereas cluster two displayed a mild similarity to cluster six (Figure 2b,f, Appendix A). Calculating how closely a gene matches the cluster core/centroid, helps identify genes that might play a role in determining the expression of the cluster as a whole [31]. For cluster three and six, a gene homologous to the Arabidopsis *PIF1/PIL5* (*Ce_contig26819/CebHLH_64*) and a gene homolog to the Arabidopsis *bZIP17* (*Ce_contig46896*/*CebZIP_41*), respectively, were the closest to the core (Appendix A). These genes may have a prominent role in regulating gene expression in cluster three and six. 

This analysis successfully identified the main gene regulatory network of photosynthesis in *C. endivia* leaves, that included homologues of several known transcriptional regulators of the photosynthetic process such as *PHYB, HY5/HYH, PIFs, GLKs, GRFs*, and *SIGs* [8]. *PHYE* and *PHYA* were found in cluster three but with lower scores to core (0.83 and 0.69, respectively) (Appendix A). A prominent role of *CebHLH_64* and *CebZIP_41* as master regulators of the photosynthetic transcriptional network in *C. endivia* leaves can be hypothesized based on their closeness to the respective cluster cores.

However, the presence of a homologue of the Arabidopsis *PIF1/PIL5* as a main positive driver of gene expression in cluster three was unexpected as *PIF1/PIL5* is a key negative regulator of phytochrome-mediated responses, and acts by inhibiting chlorophyll biosynthesis [32]. In our database, another two homologues of *PIF1/PIL5* are present in cluster six and cluster eight, indicating that they may have retained the function of Arabidopsis *PIF1/PIL5*, whereas *CebHLH_64* may have acquired a different function in *C. endivia*. A homologue of *PIF3* was also present in cluster eight, which contains genes whose expression levels remained constant during the two years of production, although always lower in ‘Flester’ as compared with the other three cultivars. 

### 2.3. Analysis of Gene Co-expression Networks of C. endivia Leaves

To provide a global overview of co-expression patterns of the *C. endivia* TF-PHOTO genes, we used a total of 23 RNA-seq samples from the two independent field experiments to construct a targeted gene co-expression network (GCN). The workflow used to obtain and analyze the GCN is shown in Figure 3.

We performed pairwise correlation analysis on TF-PHOTO gene expression values, and used significant and very strong positive and negative correlations (r ≥ |0.8|) to construct the network (Appendix A), that was then visualized and explored with Cytoscape [33] (Figure 4). Out of 1833 TF-PHOTO genes, 1406 formed a major tight network with a clustering coefficient of 0.542 and a characteristic path length of three (Figure 4a). The clustering coefficient is a measure of the degree to which nodes in a graph tend to cluster together and varies from zero to one, with zero meaning no connections and one meaning all possible connections [33]. The characteristic path length is a measure of the efficiency of information flow on a network that indicates the average number of steps along the shortest paths for all possible pairs of network nodes, the lower the number the easier the communication [33]. According to these parameters, our network is formed by highly interconnected genes that easily communicate information to each other. 

In graph theory, centrality provides a means of ranking nodes based on network structure. To identify key components (hubs) in the network, we ranked the genes according to the following three important centrality parameters: (i) node degree, which is the number of neighbors to which a node directly connects; (ii) closeness centrality, which estimates how fast the flow of information would be through a given node to other nodes; and (iii) betweenness centrality, which defines the number of information streams passing through a given node [34]. Using these centrality parameters, 10 top-ranked hub genes (highlighted in pink in Appendix A and Appendix A) were identified as those with percentile ranking of 10 (top 10%) of node degree, betweenness centrality and closeness centrality indexes. All the 10 hub genes belonged to either cluster three or cluster six, and included homologues of known regulators of photosynthesis such as sigma factors and cytokinin response regulators that mediate light effect on chlorophyll concentration [35]. Interestingly, three homologues of Arabidopsis *ETHYLENE-RESPONSIVE TRANSCRIPTION FACTORS (ERFs)* (*Ce_contig59871/ERF_80; Ce_contig147/ERF_1; Ce_contig85235/ERF_106*), one of *AUXIN RESPONSE FACTOR 9 (Ce_contig36168/ARF_9)* and one of the *MANNOSE-6-PHOSPHATE ISOMERASE 1 (Ce_contig50113/PMI2)* involved in ascorbic acid biosynthesis, resulted to play a central role in the photosynthesis transcriptional regulatory network and constitute novel potential players in this process. Within the GCN, a homologue of the Arabidopsis *MYB73* TF *(Ce_contig73456/MYB_58)* had the highest number of connections and can be also considered an important hub in the network (Appendix A). Figure 4b shows the GCN subnetwork formed by cluster three and cluster six genes, which constitutes the core of the TF-PHOTO main global network. The antagonistic relationship between genes of cluster three and those of cluster six is central in the network, and mildly influences also genes belonging to cluster one, two and seven. This is shown by the medium-high degrees of centroid-centroid correlation of cluster one, two and seven with cluster three and six, and by the existence of two main anticorrelated blocks of TF-PHOTO transcript expression (Appendix A).

Overall, GCN data were consistent with the cluster analysis and pointed to the existence of a main photosynthesis-driven TF regulatory network that plays a central role in the leaf transcriptome. 

### 2.4. Interplay Between Shape and Function: Photosynthetic Parameters in Relation to Morphological Leaf Diversity

*C. endivia* escaroles and endives represent a good model to study the relationship between leaf shape and photosynthetic activity as they are characterized by a high degree of intraspecific morphological variation (Figure 5i). 

The ‘Confiance’ and ‘Flester’ escaroles have broad leaves as compared with ‘Domari’ and ‘Myrna’ endives that develop narrow crisp curly leaves (Figure 5j). Differences between escarole and endive development are visible since the cotyledonary stage, as cotyledons are large and rounded in ‘Confiance’ and ‘Flester’ (Figure 5c,d) and more elongated and narrow in ‘Domari’ and ‘Myrna’ (Figure 5a,b). The leaves of endives are elongated and develop serrations from the first leaf onwards, whose number exponentially increases in subsequent leaves (Figure 5e,f), until they become deeply lobed or even dissected at the mature stage (Figure 5j). Differently, leaves of escaroles are rounded and smooth, with few small protrusions, and develop into large leaves with slightly wavy margins (Figure 5g,h,j). 

We measured morphological parameters of ‘Domari’, ‘Myrna’, ‘Flester’, and ‘Confiance’ plants at early stages of development, and analyzed changes in the area exposed to incident light and in the ratio of leaf dry mass to leaf area (LMA). Although leaf shape was considerably different between endives and escaroles (leaf length and width expressed as leaf index (LI), number of serrations) (Figure 5l,m), the overall leaf area exposed to incident light did not statistically differ between the two varietal types (Figure 5k). This indicates a compensatory mechanism that allows plants with different leaf shapes to balance their leaf growth rate to optimize interception of light for photosynthesis. Only ‘Flester’ differed in the LMA parameter, which is a measure of the leaf-level cost of light interception that correlates growth with carbon gain (Figure 5m), suggesting possible differences in leaf structure and anatomy [36]. 

To evaluate the photosynthetic performance of ‘Domari’, ‘Myrna’, ‘Flester’, and ‘Confiance’, measurements of chlorophyll fluorescence imaging, leaf chlorophyll content, and photochemical reflectance index were performed (Table 1). A representative image of chlorophyll fluorescence parameters, such as maximal quantum efficiency of PSII photochemistry (F_v_/F_m_) and quantum efficiency of PSII photochemistry (ΦPSII), in leaves of the cultivars used in this experiment is presented in Figure 6. Chlorophyll fluorescence analysis is a rapid nondestructive tool to evaluate the state of the photosynthetic apparatus [37]. Our results showed that the values F_v_/F_m_ were significantly different among the cultivars (Table 1). Nevertheless, our data highlighted that the values of the Fv/Fm ratio for all cultivars were within the 0.75–0.85 range, which indicate non-stressed conditions in the plants under study. The Φ PSII, the most useful parameter to measure the efficiency of PSII photochemistry [38], indicated higher values in ‘Confiance’ and ‘Domari’ with respect to ‘Myrna’ and ‘Flester’ (Table 1). A similar trend was also detectable for the electron transport rate (ETR) data, confirming the higher photosynthetic performance of ‘Confiance’ and ‘Domari’. ‘Flester’ had the lowest values of ETR as compared with the other cultivars. In addition, leaf chlorophyll content resulted higher in ‘Confiance’ and ‘Domari’ leaves but significantly lower in ‘Flester’. Furthermore, the analysis of apparent absorptivity of the leaf surface (Abs) (Table 1) showed a trend similar to chlorophyll content, corroborating a strong correlation between the leaf absorbance and the total chlorophyll content in all the cultivars analyzed. The higher photosynthetic performance for ‘Confiance’ and ‘Domari’ was confirmed through photochemical reflectance index (PRI) measurements, where PRI is a spectral index linked to photosynthetic parameters and pigments content in plant [39]. ‘Flester’ plants showed a significant lower PRI index (Table 1).

In order to study the relationship between the leaf shape and the photosynthetic activity and to evaluate the spatial heterogeneity of the photosynthetic activity on the leaf surface, on the chlorophyll fluorescence images (Figure 6), four area of interested (AOI), two in the central part of the leaf (internal) and two in the outer zones (external), were selected. Data reported in Appendix A show that, inside of each cultivar, no difference between values of F_v_/F_m_ and Φ PSII measured in the internal and external areas was found. These findings suggest that, in our experimental conditions, leaf shape did not modify the photosynthetic efficiency over the leaf surface. 

In order to evaluate the response of ‘Flester’ cultivar regarding its reduced photosynthetic performance we also measured carotenoid and malondialdehyde (MDA) contents (Appendix A). Carotenoids act as light harvesters, quenchers and scavengers of triplet state chlorophylls, and singlet oxygen species, dissipators of excess harmful energy during stress condition and membrane stabilizers [40]. MDA is produced from the lipid peroxidation of polyunsaturated fatty acid (PUFAs) in response to oxidative stress by ROS attack [41] but it has recently been suggested that MDA may act as a protection mechanism rather than being a damage indicator if the elimination of MDA and redox signaling regulation works correctly [41]. The ‘Flester’ cultivar showed a significant reduction of carotenoids and increased amount of MDA (Appendix A). These results are consistent with the physiological parameters and the altered expression of genes related to photosynthesis observed in the ‘Flester’ cultivar (Figure 7). In conclusion, in our experimental conditions the major alterations in the photosynthetic properties were due to the different cultivar genotypes, whereas no significant differences were linked to the two morphological phenotypes of endives and escaroles.

### 2.5. Interplay Between Development and Function: Developmental Genes Related to the Photosynthetic Regulatory Pathway 

In order to explore a possible link between the photosynthetic regulatory network and leaf development, we searched the cluster three and six modules for genes involved in developmental processes (GO:0044767). We found a total of 191 genes, of which 110 were in cluster three and 81 were in cluster six. Amongst developmental genes, we identified those that were positioned very close to the clusters core (score to core close to one) (Table 2). 

These included known photosynthesis regulators (bold in Table 2) such as the *GRAS* genes *GAI/RGA* (*Ce_contig18469/CeGRAS_16*) and *RGAL3 (Ce_contig31926/CeGRAS_23*) known to regulate the gibberellins signaling to repress photomorphogenesis in response to light, *PHYTOCHROME B* (*Ce_contig13510/CePHYB*) and the phytochrome A signal transduction *GRAS* gene *PAT1 (Ce_contig5939/CeGRAS_6)*, *RESPONSE REGULATOR 12 (Ce_contig68433/CeARR-B_7)* that mediates cytokinin response to light stress, *GROWTH-REGULATING FACTOR 5 (GRF5) (Ce_contig71598/CeGRF_8)* and *SIGMA FACTOR 6 (CeSIG6) (Ce_contig85637/CeSIG_5)* involved in the chloroplast response; four homologs of *ZINC FINGER PROTEIN 10 (Ce_contig18795/CeC2H2_2,; Ce_contig46000/CeC2H2_40, Ce_contig76425/CeC2H2_62*, and *Ce_contig81969/CeC2H2_71)*. Interestingly, several genes related to cell proliferation and leaf morphology, which were not previously linked to photosynthesis and light response, were strictly connected to the photosynthesis network (Table 2). These genes include homologues of the Arabidopsis *LOST MERISTEMS 3* (*Ce_contig82634/CeGRAS_43*; Ce_contig6575/GRAS_7); *TEOSINTE BRANCHED, CYCLOIDEA AND PCF 14* and *20 (Ce_contig33120/CeTCP_7; Ce_contig49086/CeTCP_12)*; *MONOPTEROS (Ce_contig38256/CeARF12), MYB DOMAIN PROTEIN 17/LATE MERISTEM IDENTITY2 (Ce_contig1183/CeMYB_2), ATHB8 (Ce_contig83059/CeHD-ZIP_41*), N*AC DOMAIN CONTAINING PROTEIN 83 (NAC083)/VND-INTERACTING 2 (Ce_contig47321/CeNAC_49), ASYMMETRIC LEAVES 2-LIKE 1 (Ce_contig19510/CeLBD_6)* and *BEL1-LIKE HOMEODOMAIN 8 (Ce_contig85206/CeTALE_22)*. Overall, cluster three was enriched in positive regulators of photosynthesis, whereas cluster six was enriched in redox responsive and photooxidative stress genes. 

### 2.6. TF-PHOTO Genes Differentially Expressed in Broad Versus Curly Leaves 

We next examined genes that were always significantly differentially expressed between broad leaves escaroles (‘Flester’ and ‘Confiance’) and curly leaves endives (‘Domari’ and ‘Myrna’) over the two production years. Only three transcription factors were found invariably differed in their expression between endives and escaroles, which were homologous to the Arabidopsis TFs *BREVIS RADIX-*like *3 (Ce_contig84656/CeBRX_1), ETHYLENE-RESPONSIVE TRANSCRIPTION FACTOR RAP2-12 (Ce_contig50460/CeERF_72),* and *MYB113 (Ce_contig5865/CeMYB_77)*, which were placed in cluster five, eight and three, respectively. The differential expression of these genes was confirmed by qRT-PCR in mature leaves. Amongst the three genes, *CeMYB_77* and *CeERF_72* were reconfirmed as differentially expressed in younger leaves, whereas the expression of *CeBRX_1* was not detectable in young leaves in our qPCR analysis (Figure 8). 

## 3. Discussion

The leaf shape adaptive value in relation to its photosynthetic activity represents a standing question that is both of fundamental and applied interest. We addressed this issue in *C. endivia* (escaroles and endives), a typical Mediterranean species of the Asteraceae (or Compositae) that has been cultivated since ancient times in the Mediterranean basin. *C. endivia* is a close relative of *Lactuca sativa*, a crop model species for the Asteraceae that represents one of the largest and most successful flowering plant family [28]. Due to the striking morphological diversity of their leaves, escaroles and endives represent a good model to study the relationship between leaf shape and photosynthetic activity.

### 3.1. The Regulatory Network of Photosynthesis in the Leaves of C. endivia

We took advantage of an in-house produced transcriptome [25] and NGS leaf gene expression datasets to extract transcription profiles of leaf TFs and photosynthesis-related genes to explore regulatory networks in the leaves at the system level. To this purpose, transcriptome and RNA-seq data were deeply refined to obtain a reliable non-redundant gene expression database (TF-PHOTO) suitable for cluster and gene co-expression analysis. Clustering gene expression data allows to identify substructures in the data, and identify groups of genes that may share a biological function or be under the same transcriptional control (co-regulated) [31]. GCN are graphical representations of complex interactions where genes are represented by nodes, and edges connect genes that are significantly co-expressed [34]. Cluster analysis and GCN approaches successful identified the photosynthetic transcriptional regulatory network occurring in leaves and the candidate hub genes that may rule the photosynthesis transcriptional network. The identified regulatory network involved known genes of two antagonistic pathways. one concerning light signal transduction to promote the expression of photosynthesis master regulators and activate chloroplast response and downstream antenna, photosystem, and rubisco gene expressions (genes in cluster three). The other related to photooxidative stress, RS, and unfolded protein response (UPR) (genes in cluster six). This is consistent with the large increase in the production of reactive oxygen species (ROS) derived from light-driven energy transfer and electron transport during the photosynthetic process [5]. ROS are key components of chloroplast-nucleus RS pathways and also link to UPR through specific plastid signals [42]. Redox changes can influence the susceptibility of photosynthesis to high-light-induced inhibition and also modulate chloroplast-to-nucleus signaling and the transcriptional responses of leaves to high light [5]. Cluster analysis pointed to a major role of the *AtPIF1/PIL5* homologue *CebHLH_64* and the *AtbZIP17* homologue *CebZIP_41* in determining the characteristic gene expression profiles in cluster three and cluster six, respectively, based on their closeness to the respective cluster cores. This finding is in contrast with the known function of AtPIF1/PIL5 as a negative regulator of PHYB-induced light response and chloroplast biogenesis. However, as other two *PIF1/PIL5* homologues are present in cluster six and eight, this member may represent an example of gene neofunctionalization, or its expression profile may not correspond to its protein activity that, in Arabidopsis, is regulated by phytochrome-induced rapid degradation [43]. The Arabidopsis bZIP17 is a membrane-associated basic-leucine zipper TFs that resides in the endoplasmic reticulum (ER). It was shown to act as a plant stress sensor/transducer that in response to stress is activated by proteolytic release of the N-terminal from the ER membranes allowing its translocation to the nucleus [44]. It is tempting to speculate that CebZIP_41 might have a major role in transducing the ROS signal derived from the photosynthesis-induced photooxidative stress to activate the ROS-mediated transduction network [45].

Network topology is the layout of nodes and edges, and the topological properties determine the functional aspects of the relationships [34]. Parameters derived from network local properties are commonly used for node ranking to identify essential genes in the network [46]. Topological analysis of the whole TF-PHOTO GCN identified several candidate hub genes that may have a central role in regulating the leaf transcriptome as a whole. These genes belong to either cluster three or cluster six, indicating a prominent role of the photosynthetic function in the global leaf gene expression (yellow nodes in Figure 9).

Four of them are homologous to known regulators of photosynthesis and include: *AtARR12*, which mediates cytokinin response to light; the sigma factor *SIGA* and the sigma factor *SIGF*, which positively regulate chloroplast response and biogenesis; and *AtABF2*, which is involved in the ABA-mediated chlorophyll degradation and leaf senescence [47]. However, the *ABF2* homologue *Ce_contig73259/CebZIP_64* belonged to cluster three and was highly co-expressed with photosynthesis-induced downstream genes and was downregulated in the ‘Flester’ cultivar. This finding may underlie a different function of this homologue in *C. endivia* leaves. Regarding the other identified hubs, three ethylene responsive factors (ERFs) were amongst the top-ranked genes. ERFs act as critical downstream components of the ethylene signaling in regulating plant development and stress responses, but little is known about their role in regulation of photosynthesis [48]. Interestingly, overexpression of *HARDY*, an AP2/ERF gene from Arabidopsis, showed enhanced drought tolerance in rice by a reduction in transpiration and improvement of photosynthesis, probably due to an increase in leaf biomass and bundle sheath cells [49]. Differently, the rice *AP2/ERF-N22* gene is able, when overexpressed, to increase carotenoid levels and reduce chlorophyll content, thus, leading to reduced photosynthetic rate and efficiency [50]. A few *ERF* genes were shown to promote chlorophyll degradation during senescence [51,52], whereas overexpression of the citrus *CitERF13* gene in *N. tabacum* leaves resulted in a significant decrease in net photosynthetic rate and of Fv/Fm ratio [53]. These data indicate contrasting roles of ERFs in the photosynthetic pathway and are consistent with our data that place two *ERFs, Ce_contig147/ERF_1* and *Ce_contig85235/ERF_106*, in cluster six, and *Ce_contig59871/ERF_80* in cluster three. A homologue of the *AUXIN RESPONSE FACTOR 9 Ce_contig36168/CeARF_9*, is also amongst the 10 top-ranked genes in the GCN (Figure 9). *CeARF_9*, together with three other *ARFs*, co-expressed with many different important regulators of the photosynthetic network, such as the Arabidopsis homologues of *PHYB*, *ABF2*, *ARR12*, as well as several *SIG* and *GAI*, highlighting a prominent role of auxin response in integrating signals from different pathways into a developmental frame. 

Overall, the identification of several hormone response genes as potential hubs in the TF-PHOTO regulatory network points to a major role of hormone signaling in integrating the antagonistic transcriptional responses related to the photosynthetic function of the leaf. In addition to the 10 top-ranked genes, the gene *CeMYB_58*, placed in cluster six, showed the highest number of connections in the GCN and can also be considered a major hub. *MYB* genes are involved in many aspects of plant development and in the response to environmental cues [54,55,56]. The analysis of the amino acid sequence of CeMYB_58 showed a high similarity with the Arabidopsis MYB TFs belonging to the subgroup 22, namely, MYB73, MYB70, MYB44, and MYB77. These MYBs seem to be involved in the activation of different pathways related to plant hormone signaling and stress response [57], but their role in photosynthesis-related processes is completely unknown. *CeMYB_58* is strongly positively correlated with *CebZIP_41/AtbZIP17* and *CePMI2/AtPMI1*, and strongly anticorrelated to genes encoding subunits of the chloroplast NAD(P)H dehydrogenase complex and a violaxanthin deepoxidase (*VDE*). VDE catalyses the conversion of violaxanthin to zeaxanthin through de-epoxidation [22] and is a key step of the xanthophyll cycle, which largely contributes to nonphotochemical quenching (NPQ) to avoid photoinhibition, occurring when the light energy absorbed by plant leaves exceeds its consumption, through the dissipation of excessive energy as heat [58]. Moreover, VDE is involved in ABA biosynthesis [22], then it may have a role in integrating the hormone signaling and the photosynthetic function of the leaf. A role of *CeMYB_58* in the negative regulation of *VDE* is plausible.

Gene expression analysis at the leaf system level, as well as physiological and biochemical analysis, failed to identify differences in the photosynthetic regulatory networks and photosynthetic performance related to the different leaf shapes of endives and escaroles. Leaf morphology did not seem to influence photosynthetic properties in the *C. endivia* genotypes here analyzed, and the differences observed in the the endive and escaroles cultivars were genotype dependent. However, the robustness of our cluster and GCN analyses allowed us to investigate the relationship between photosynthesis and leaf development genes under the so-called “guilt by association” rule, which provides a powerful framework to associate functions within a gene expression module based on correlated gene expression pathways [47]. We identified genes involved in the developmental processes as strictly associated to the photosynthetic regulatory clusters. Among them, several were already described to participate in light response processes, whereas other have not been previously associated to photosynthesis-related pathways. The latter may represent novel players in the interplay between leaf development and photosynthesis and will be described in the next paragraphs.

### 3.2. Developmental Genes in Cluster Three Linked to Positive Regulation of Photosynthesis

Cluster three contains several homologues of Arabidopsis known developmental genes that participate in light response processes such as *PHYB*, the *GRAS* genes *GAI* and *RGL3, ARR12, GRF5, SIG6,* and *MYB17* (Table 2) [8]. GRAS proteins play important roles in plant growth and development ranging from GA signaling, light signal transduction, and axillary shoot meristem formation [59]. In cluster three, in addition to *GAI* and *RGL3*, we found two homologues of Arabidopsis *LOST MERISTEMS 3* (*LOM3/SCL6-IV*) (Ce*GRAS_43* and *CeGRAS_7* in Table 2), which play vital roles in the proliferation of meristematic cells promoting shoot indeterminacy via a non-cell-autonomous pathway [60]. Moreover, three scarecrow-like (SCL) proteins, including AtSCL6-IV, were shown to be targeted by miR171 to negatively regulate chlorophyll biosynthesis [61] and may link chloroplast development and cell proliferation. *CeGRAS_43* expression was highly correlated with genes encoding thylakoid membrane-associated proteins like THYLAKOID RHODANESE-LIKE (TROL) and PSB photosystem II subunits. Interestingly, *CeGRAS_43* expression was also highly correlated to *CeGRF5*, which plays a major role in stimulating chloroplast division and positively affects leaf size when overexpressed [62]. *CeARF_11*, a homologue of Arabidopsis *ARF16*, was also co-expressed with several photosynthetic downstream genes and with one of the top-ranked genes in the GCN, *CeARF_9*. Auxin response regulators have been recently shown to play antagonistic roles in photosynthesis regulation in tomato, and the *SlARF6A* gene was shown to positively affects photosynthesis in fruits and leaves of tomato plants [63]. Our data are consistent with a major role of auxin response in coordinating photosynthesis and leaf development. 

Three *MADS box* homologous to the Arabidopsis *SVP, MAF3*, and *AGL62* were also in cluster three. SVP is a well known regulator of flowering transition in response to light and ambient temperature, however, was also proposed to affect steady-state photosynthetic rates [64]. Amongst these genes, the *C. endivia* homologue of *MAF3* (*CeMIKC-MADS_13*) was highly connected in the GCN and was strongly co-expressed with several photosynthetic genes and the GCN hub *CeERF_80*.

Two homologues of the Arabidopsis *TEOSINTE BRANCHED, CYCLOIDEA AND PCF 14* and *20 (CeTCP_7; CeTCP_12)* were found in cluster three. Members of the Arabidopsis TCP family act on several aspects of plant development [65]. *CeTCP_7*, homologous of *TCP14*, was highly connected in the GCN and with the GCN hub *CeERF_80* and is part of the same regulatory subnetwork. *AtTCP20* is involved in the control of multiple antagonistic processes during leaf growth by linking regulation of growth and cell division via the jasmonic acid (JA) pathway [66]. In developing leaves, TCP20 can also directly control the expression of genes involved in iron homeostasis [67] which is crucial for proper photosynthetic electron transport chain functioning and chloroplast development. However, *tcp20* knocked-out analyses did not reveal direct effects on photosynthetic efficiency [68], and suggested a role as a negative regulator of leaf aging [69]. The endive homologue of *TCP20, CeTCP_12*, had a few strong links in the GCN but was highly correlated with a homologue of the Arabidopsis *ABF2, CebZIP_64*. However, *CebZIP_64* involvement in photosynthesis in *C. endivia* seems opposite to that observed for *ABF2* that mediates ABA-dependent chlorophyll degradation and aging in Arabidopsis [47]. 

The Arabidopsis ATHB8 is a known regulator of vascular development [70]. The *ATHB8*-like gene of endive (*CeHD-ZIP_41*) shares maximal sequence identity with the *HOX32* gene of rice, which was suggested to play a key role in the interplay between leaf development and proper photosynthetic machinery functioning [71].

### 3.3. Developmental Genes in Cluster Six Antagonistic to Photosynthesis

Cluster six was enriched in genes related to the photooxidative stress and ROS production that derived from light-driven energy transfer and electron transport during the photosynthetic process [5]. Chloroplast redox homeostasis and ROS are key components of chloroplast-nucleus retrograde signaling pathways [72] and also link to UPR through specific plastid signals [73]. RS is a complex intracellular signaling pathway that chloroplast utilizes to convey information on their physiological states to the nucleus and modulate the expression of nuclear genes accordingly, coupling physiological state with developmental signaling [74,75]. Recently, Martin et al. [18] showed that the phytochrome and RS pathways converge antagonistically to regulate the expression of *GLK1*, a key regulator of a light-induced transcriptional network central to photomorphogenesis.

Cluster six harbors several genes involved in different developmental processes. Among these, we found an homologue of the Arabidopsis *REDOX RESPONSIVE TRANSCRIPTION FACTOR 1 (RRTF1) (CeERF_75)*, a central TF in fast retrograde signaling to high light response [76,77]. *CeERF_75* was highly connected in the GCN, with very high correlation with several TFs (including other *ERF* members) confirming the central role of RRTF1-like proteins in the redox responsive regulatory co-expression network acting during RS [77].

Cluster six also contained four homologues of the Arabidopsis *SALT TOLERANCE ZING FINGER* (*ZAT10*) (Table 2), which is involved in development and growth in response to oxidative stress. Constitutive expression of *ZAT10* elevates the expression of reactive oxygen defence transcripts and results in palisade tissue formation [78,79]. The four endive homologues were highly co-expressed and tightly connected to the GCN. Specifically, *CeH2C2_71* established 412 connections and can be considered a main hub in the oxidative stress response. Intriguingly, *CeH2C2_71* and *CeERF_75* were strongly co-expressed, suggesting a crosstalk between ZAT10 and the RRTF1-mediated RS pathway. *CeNAC_1* and *CeNAC_29*, homologous to the Arabidopsis *NAC053* and *VNI2*, were also part of the same major *CeH2C2_71* and *CeERF_75* network and may link oxidative response to activation of transcription factors involved in leaf senescence. NAC053 is thought to promote ROS production by binding directly to the promoters of genes encoding ROS biosynthetic enzymes during drought-induced leaf senescence in a network involving WRKY genes [80], whereas VNI2 is involved in vascular development and plays a role in the age-dependent induction of stress resistance [81]. Several *WRKY* homologous to Arabidopsis *WRKY22, 53, 75* were found in cluster six. WRKY22, *WRKY53* and *WRKY75* are involved in leaf senescence. *WRKY22* participates in the dark-induced senescence signal transduction pathway [82], and its transcription is suppressed by light and promoted by darkness and by H_2_O_2_. *WRKY53* play a central role in senescence regulation [83,84] and in the crosstalk between early and late stages of leaf development [85]. Three endive homologues, *CeWRKY_25, 29*, and *34*, were strongly co-expressed and highly connected in the GCN, and co-expressed with *HEAT SHOCK FACTOR* TFs and *CebZIP_41/AtbZIP17*, a major player in the unfolded protein response in the ER that we also identified as a main transcriptional driver in cluster six. 

Furthermore, *MYB* genes were also found in cluster six, in particular the homologues of Arabidopsis *MYB14* (*CeMYB_9*) and *MYB102* (Ce*MYB_36*). MYB102 is involved in the integration of wounding and osmotic stress signals [86] and it is a possible interactor of the SWI/SNF chromatin remodelling ATPase BRAHMA, which promotes leaf growth by stage-specific modification of CK responses [87]. 

A TALE homeobox homologous to the Arabidopsis *BLH8* (*CeTALE_22*) was placed in cluster six. TALEs are encoded by two small subfamilies in the Arabidopsis, *KNOX* and *BLH* [88]. TALEs are master controllers of plant development that regulate the fine equilibrium between cell fate maintenance and differentiation in the lateral organ formation at the shoot apical meristem (SAM) and during leaf morphogenesis and leaf marginal outgrowth [88,89]. In Arabidopsis, BLH8 interact with the class 1 KNOX SHOOT MERISTEMLESS and BLH9 to restrict organogenesis at the SAM [90]. In endive leaves, the expression of *CeTALE_22* was highly correlated with the expression of *CeLBD_6*, the endive homologue of *ASL1*, which, in Arabidopsis, seems to be involved in the transcriptional regulation of class 1 *KNOX* during lateral organ differentiation [91]. Interestingly, members of class I KNOX were found to act as transcriptional activators of *SlGLK2* in tomato to promote chloroplast [92], whereas a BLH, SlBEL11, was shown to play an important role in chlorophyll synthesis and chloroplast development in tomato fruits [93].

### 3.4. Transcription Factors that Are Differentially Expressed Between Broad- and Curly-Leaved C. endivia Cultivars

Finally, three genes that are differentially expressed between broad and curly leaves were identified and will be further investigated to assess their role in leaf development and shape.

Among them, *CeMYB_77* belonged to the photosynthesis regulatory cluster thee. The amino acid sequence of *CeMYB_77* is highly similar to those of the Arabidopsis genes *AtMYB113/PAP3* and *AtMYB75/PAP1*, involved in anthocyanin biosynthesis and accumulation [13,94]. Interestingly, AtMYB75/PAP1 is involved in a light-dependent signaling pathway. JA promotes anthocyanin accumulation under far-red light through a signaling pathway consisting of PHYA, COP1, and MYB75/PAP1 [95]. Moreover, MYB75/PAP1 physically interacts with the KNOX transcription factor KNAT7, to form functional complexes that contribute to the regulation of secondary cell wall deposition and that integrate the metabolic flux through the lignin, flavonoid, and polysaccharide pathways in Arabidopsis [96]. Thus, a role of *CeMYB_77* in light-mediated anthocyanin production in *C. endivia* can be hypothesized.

The other two DEGs, *CeERF_72* and *CeBRX_1*, homologous to *AtRAP2-12* and *AtBRX3*, were found in cluster eight and five, respectively. At*RAP2.12* exerts a major control in the metabolic reprogramming of cells under hypoxic conditions [97,98], but a specific function in leaf development or shape has not been described. In Arabidopsis, BRX proteins form a small family that was shown to control the extent of cell proliferation and elongation in the growth zone of the root tip [99]. BRX proteins are highly similar, contain four highly conserved domains of unknown function, and a specific role in leaves has been so far described. 

The main regulatory pathways involved in leaf shape determination and leaf margin serrations in Arabidopsis and *C. irsuta*, *CUP-SHAPED COTYLEDON2* (*CUC2*) boundary genes, *SHOOT MERISTEMLESS* (*STM*), and *REDUCED COMPLEXITY* (*RCO*) [100,101] were not expressed in mature leaves and their role in determining leaf shape in *C. endivia* could not be addressed at this stage. 

## 4. Materials and Methods 

### 4.1. Biological Material and Morphological Analysis

*Cichorium endivia* L. curled-leaved endives ‘Domari’ and ‘Myrna’ (*C. endivia* var. *crispum*), and broad-leaved escaroles ‘Confiance’ and ‘Flester’ (*C. endivia* var. *latifolium*) seeds were provided by the Enza Zaden company (www.enzazaden.com). Plants used for transcriptomic analysis were first grown in a nursery, then three-week-old seedlings were moved into an open field (8.2 plants/m^2^) and harvest occurred during the second half of November (86 days after sowing) as described in Testone et al. [25]. Two production cycles were carried out in 2011 and 2012, in the same location and in the same time of the year. 

Seeds were sown in water wet paper in Petri dishes and germinated in the light in a growth chamber at 23 °C under long-day conditions (16 h/8 h light/dark at 150 µmol). After one week from germination, plantlets were transferred in soil pots, watered every two days and nutritive solution added once a week. For photosynthesis measurements, 21-day-old plantlets were placed in bigger pots in the soil and transferred to a greenhouse. The experiment was set up in a completely randomized design with at least three replicates (pots) for each of the four cultivars. Pots were placed during summertime (July to August) in a greenhouse under natural photoperiod (about 14 h), with mean (night and day) temperatures of 22.1 to 28.8 °C and relative humidity of 50% to 60%. Plants were irrigated daily by supplying the water loss for evapotranspiration to maintain 50% of the water holding capacity, evaluated before starting the experiment.

For morphological analysis, leaf length, leaf width, and number of serrations were recorded in 10 plants per each cultivar from 10 to 21 days after germination. For calculating the leaf area exposed to incident light, pictures were taken, and areas were calculated using the imaging software NIS-Elements-BR Ver. 2.1 (Nikon). 

### 4.2. RNA Isolation, Sequencing, and Expression Analysis

For the RNA-seq analysis of the *C. endivia* production cycle 2011, leaf sampling (three biological replicates per cultivar), total RNA extraction and quality assessment, cDNA libraries synthesis, and Illumina sequencing were as described in Testone et al. [25]. Raw sequenced reads were checked and filtered as previously described [25], and aligned to the reference transcriptome using Kallisto [102]. Digital gene transcription levels were expressed in RPKM (reads per kilobase per million mapped reads) values.

For real-time RT-PCR (qRT-PCR) gene expression analysis, total RNA was isolated (TRIzol, Invitrogen), purified (Plant RNeasy RNA Cleanup, Qiagen, Hilden, Germany), and treated with DNase I (Qiagen). First-strand cDNA was synthesized from 1.5 µg of total RNA using the Superscript III first-strand synthesis system (Invitrogen). Primer design was performed using Primer3 software [103], followed by primer test specificity with the Primer-Blast NCBI tool in the Cichorioideae database. Quantitative RT-PCR reactions were performed using the Eco Real-Time PCR System (Illumina), following manufacturer’s instructions using 50 to 70 ng of template cDNA (Biotool easy mix with Eva green fluorophor) and 300 nM final primer concentration. Cycling conditions were as follows: 95 °C for 10 min, 50 cycles at 95 °C for 5 s, 68 °C for 15 s, and 72 °C for 15 s. The experiments included at least two biological and three instrumental replicates. Gene expressions were normalized against the ACTIN reference gene [104] and mean normalized expressions were calculated using the Q-Gene program [105]. The primers used are listed in Appendix A. For confirmation of RNA-seq data, the purified total RNA used for the transcriptomic analysis was used. For gene expression analysis in young leaves, total RNA was extracted from pools of leaves (three young fully expanded leaves from three different 30-day-old plants) of the four *C. endivia* cultivars (two biological replicates). 

### 4.3. TF-PHOTO Database Construction

The transcriptome of *C. endivia* (cv ‘Domari’; NCBI TSA accession: GGQM00000000.1) obtained in our lab [25] was used to construct the TF-PHOTO database. The transcription factors previously identified based on the PlantTFDB database (Jin et al. NAR 2014) were used for further refinements. Contigs homologous to the *L. sativa* TFs annotated in PlantTFDB or published [27] were searched in the *C. endivia* transcriptome and added to the initial TF database if not present. To obtain reliable non-redundant full-length and near full-length TF transcripts (transcript completeness ≥ 0.7), sequences were aligned to the *L. sativa* genome [28] (1 hint/locus) and full-length percentage prediction FL% ≥ 0.7, calculated as the length of *C. endivia* deduced protein/*L. sativa* annotated protein. Predictions were further validate with the MapMan package [29] by binning of Arabidospis IDs of the corresponding TFs to MapMan bins by the Mercator application [106]. Finally, only transcript expressed in leaves (genes with read count < 10 in more than 90% of the samples, following WGCNA platform suggestions [107] in RNA-seq leaf data (NCBI Bioproject PRJNA417356) were considered. This pipeline allowed to reduce the initial 5476 predicted TFs to 1560 bona fide leaf TFs. 

As for photosynthesis and photosynthesis-related genes (energy dissipation and reactive oxygen species detoxification), the selection criteria included: (i) homology with Arabidopsis genes belonging to GO:0015979 (photosynthesis), GO:0009813 (flavonoid metabolic processes), GO:0016116 (carotenoid metabolic process), and GO:0019852 (ascorbic acid metabolic process) terms; (ii) transcript completeness ≥ 0.7; and (iii) leaf expression occurrence. All the selected transcripts were aligned to the latest release (GenBank accession GCF_002870075.1) of the *L. sativa* reference genome [28] by GMAP [108] (Appendix A).

### 4.4. K-Means Cluster Analysis and Gene Co-Expression Network Construction

For the K-means cluster analysis, we used RPKM means of the TF-PHOTO genes of endives and escaroles from leaf RNA-seq of plants grown in two independent production cycles (from 2011 to 2012). The data were log-transformed using log2(x + 1) for normalization and scaled. To determine the optimum number of clusters, the sum of squared error (SSE), the average silhouette width, and the Calinski–Harabasz index, based on the intra- and intercluster sum of squares, methods were used. The pipeline of the analysis is shown in Appendix A. Eventually, we empirically established the optimum number at 8 using a posteriori validation that consists in correlating cluster centroids to each other and then choose the highest number of clusters with centroids maximum correlation below 0.8. Data scaling, K-means clustering, and visualization were performed in R. 

For GCN construction and analysis, we used RPKM from biological replicates in the four cultivars for two production cycles (23 RNA-seq samples in total). The data were log-transformed using log2(x + 1) for normalization and Pearson pairwise correlation analysis was conducted across the selected samples using the “corrplot” and “hclust” packages of R software [109]. Significant correlations (*p* value ≤ 0.05) with a Pearson’s correlation coefficients (r) ≥ |0.8| were used for the construction of co-expression networks and network analysis in the Cytoscape software platform v. 3.5.1 [33]. 

### 4.5. Chlorophyll Fluorescence Measurements 

After 15 days of growth under greenhouse conditions, chlorophyll fluorescence measurements were performed to assess the efficiency of the photosynthetic apparatus of *C. endivia* cultivars. In particular, the maximal quantum efficiency of PSII photochemistry (F_v_/F_m_) and the quantum efficiency of PSII photochemistry (ΦPSII) were measured on two fully developed leaves per plant using a chlorophyll fluorescence imaging (MAXI-Imaging-PAM, Walz, Germany). Leaves were dark adapted for at least 30 min before determining F_0_ and F_m_ (minimum and maximum fluorescence, respectively). The F_v_/F_m_ value was calculated as (F_m_ − F_0_)/F_m_. Leaves were then adapted to a photosynthetic photon flux density (PPFD) of 370 μmol m^−2^ s^−1^ (near to the growth light intensity used in the experiment) and a saturating pulse was applied to determine the maximum fluorescence (F_m_′) and steady-state fluorescence (F_s_) during the actinic illumination. Saturation pulse images and values of the chlorophyll fluorescence parameters were captured. The ΦPSII value was calculated using the formula (F_m_’ − F_s_)/F_m_’. The apparent photosynthetic electron transport rate (ETR) was calculated as follows: ETR = ΦPSII × PPFD × 0.5 × Abs, where Abs is the apparent absorptivity of the leaf surface and 0.5 is the fraction of light absorbed by PSII antennae [110]. The Abs value was automatically calculated, pixel by pixel, from the R (red) and NI (near infrared) images using the formula, Abs = 1 − (R/NI). Moreover, the Imaging-PAM software was used to select, on the chlorophyll fluorescence images, four area of interested (AOI), two in the central part of the leaf (internal) and two in the outer zones (external), in order to evaluate the spatial heterogeneity of the photosynthetic activity on the leaf surface. 

### 4.6. Determination of Chlorophyll and Total Carotenoids Content

Measurements of total chlorophyll content were performed using the chlorophyll meter readings (SPAD-502, Minolta Camera Co., Osaka, Japan) on at least two fully developed leaves per plant. Four SPAD readings were taken from the widest portion of the leaf lamina, while avoiding major veins. The four SPAD readings were averaged to represent the SPAD value of each leaf. SPAD values were converted to chlorophyll content (µg cm^−2^) using the following equation [111]:
chlorophyll content = (99 × SPAD value)/(144 − SPAD value)


For biochemical analysis, chlorophyll-a, chlorophyll-b, and carotenoid pigments of plants (0.2 g) were extracted in 10 mL 80% chilled acetone in the dark. After centrifugation at 10,000× *g* for 10 min at 4 °C, the absorbance of the supernatants was read spectrophotometrically at 663, 647, and 470 nm. Chl-a, Chl-b, and carotenoid contents were determined according to the equations described by [112] and the results expressed in mg carotenoids per gram of FW.

### 4.7. Measurements of Spectral Reflectance and Spectral Reflectance Indices

Absolute reflectance spectra of the plants were collected using an analytical spectral device (ASD) Inc. Field Spec® 3 as reported by [113]. The spectral range of the instrument is 350 to 1025 nm with a 1.4 nm bandwidth. A white reference Spectralon panel was used to calibrate spectral measurement. An external 50 W halogen lamp was set up with an illumination angle of 45°. Besides the lamp, there were no other sources of illumination in the laboratory, and therefore no environmental contribution. The instrument fiber optic was fit through a mounting gun attached to a tripod and adjusted to a nadir viewing position. The distance from the fiber optic to the sample was 20 cm, which translated to a field of view (FOV) of approximately 90 mm (using a 25° bare fiber optic). Reflectance spectra were recorded as the ratio of sample data to white reference (99% reflectance Spectralon panel) data under the same illumination and viewing conditions. The mean of the five spectra was then determined to provide a single spectral value. Several spectral reflectance indices were derived from the collected data and are reported in Appendix A.

### 4.8. Measurement of Lipid Peroxidation Levels

Determination of lipid peroxidation was performed by measuring malondialdehyde (MDA) content following the modified method by [114]. Frozen samples were homogenized in a prechilled mortar and pestle with two volumes of ice-cold 0.1% (*w*/*v*) trichloroacetic acid (TCA) and centrifuged for 15 min at 16,000× *g*. Assay mixture containing 1 mL aliquot of supernatant and 2 mL of 0.5% (*w*/*v*) thiobarbituric acid in 20% (*w*/*v*) TCA was heated to 95 °C for 30 min and then rapidly cooled in an ice bath. After centrifugation (16,000 × g for 10 min at 4 °C), the supernatant absorbance (532 nm) was read, and values corresponding to nonspecific absorption (600 nm) were subtracted. The MDA concentration was calculated using the extinction coefficient (155 mM^−1^ cm^−1^). The lipid peroxidation levels were expressed as nanomoles of MDA per gram of FW.

### 4.9. Statistical Analysis

For differential expression analysis, the read counts matrices were normalized by TMM method [115] and processed by the edgeR package [116]. Differentially expressed genes (DEGs) were analyzed for each cultivar comparison and those with false discovery rate (FDR) ≤ 0.05 and an absolute log2 fold change ≥ 1 were considered as DEGs. 

One-way analysis of variance (ANOVA) was performed for physiological measurements, morphological and biochemical studies, and for qRT-PCR analysis. Data reported in the tables and figures refer to the average values of replicates ± standard error (SE). Normally distributed data were evaluated by employing Student–Newman–Keuls test at a significance level of *p* ≤ 0.05 using the SPSS (Chicago, IL, USA) software tool for physiological dataset while Tukey’s test with a significance of *p* < 0.05, using R statistical package vers. 3.5.0 (R Core Team, 2018) for morphological biochemical and gene expression measurements.

## 5. Conclusions

Gene expression analysis at the leaf system level, a molecular, physiological, and biochemical analysis, allowed us to investigate possible links between photosynthesis and leaf morphology in *C. endivia* cultivars characterized by broad or curly leaves morphology. Although no correlation between leaf shape and photosynthetic performance was observed, important genotype-specific differences were identified and correlated to changes in gene expression of photosynthetic genes and their regulators. The first *C. endivia* database for leaf transcription factors and photosynthesis-related genes was produced and could constitute an important resource for crop breeding and genetic improvement in Asteraceae species. Robust clustering and GCN analyses allowed us to reconstruct the regulatory pathway of photosynthesis in this important crop species. A “guilt by association” approach let us identify developmental genes strictly associated to the photosynthetic regulatory network that may thus represent novel players in the interplay between leaf development and photosynthesis. Finally, the identification of three transcription factors that are differentially expressed in *C. endivia* broad vs. curly leaves could allow us to further investigate the genetic basis of leaf morphological diversity in Asteraceae species.

## Figures and Tables

**Figure 1 plants-08-00531-f001:**
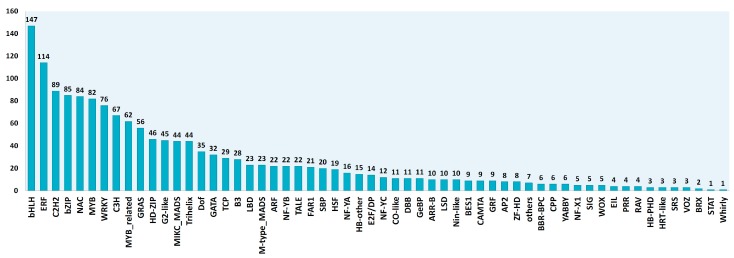
Identification and assignment of the 1560 transcription factors (TFs) expressed in the leaves of *C. endivia* according to the PlantTFDB rules [26].

**Figure 2 plants-08-00531-f002:**
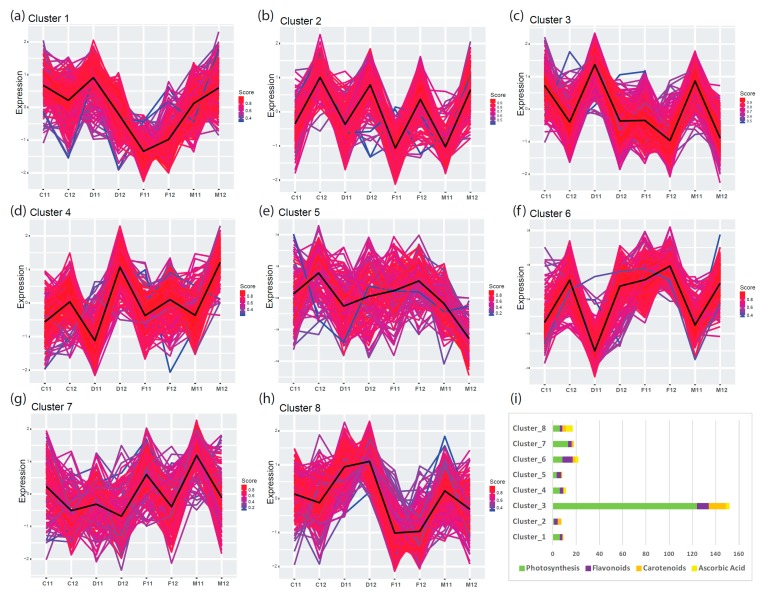
K-means cluster analysis of TF-PHOTO gene expression in the leaves of field-grown endives and escaroles for two production cycles (from 2011 to 2012). Distance matrix for k-mean clustering was calculated by Euclidean similarity measurement and using centered Pearson’s correlation as the distance metric, resulting in eight gene clusters (**a**–**h**). Genes with a profile close to the core have a score approaching one (red) while those with divergent patterns have a score closer to zero (blue). Thick lines show centroid tendencies in each cluster according to cultivar and production year. C, Confiance; D, Domari; F, Flester; and M, Myrna. Cluster three (**c**) contains most of the genes directly related to photosynthesis and is highly anticorrelated with cluster six (**f**) (r = −0.97) which likely contains negative regulators of photosynthesis or antagonistic pathways. (**i**) Graphical visualization of enrichment results and gene characteristics according to gene ontology (GO) functional categories.

**Figure 3 plants-08-00531-f003:**
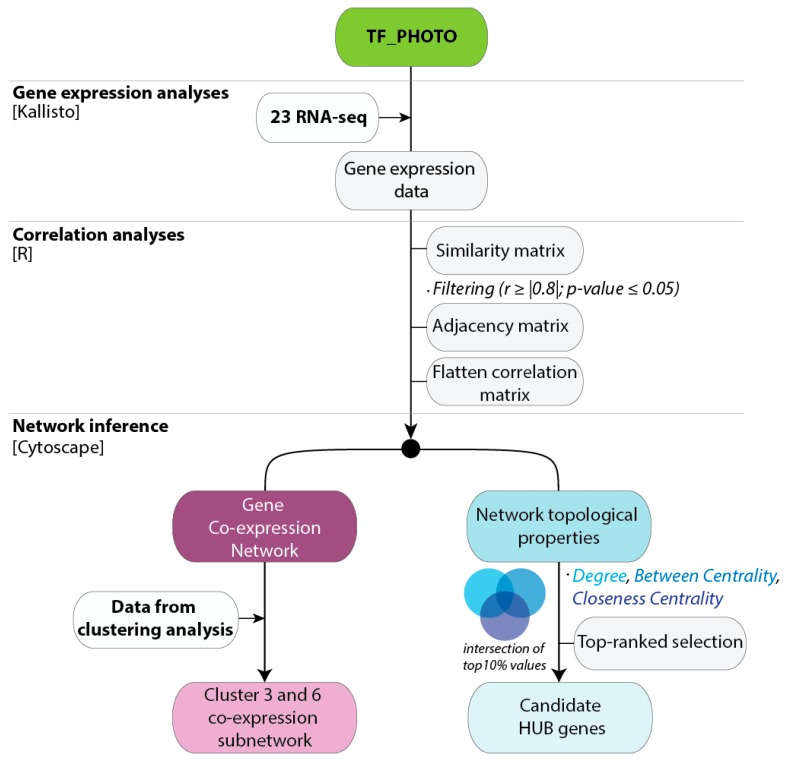
Workflow used to generate the gene co-expression Networks for the TF-PHOTO database. All the information about the database, including sequences, annotations, and the original RPKM data used for the analyses is available in Appendix A. The flatten matrices of the pairwise correlation data for r ≥ |0.8| and r ≥ |0.9| are in Appendix A and can be directly used to feed Cytoscape.

**Figure 4 plants-08-00531-f004:**
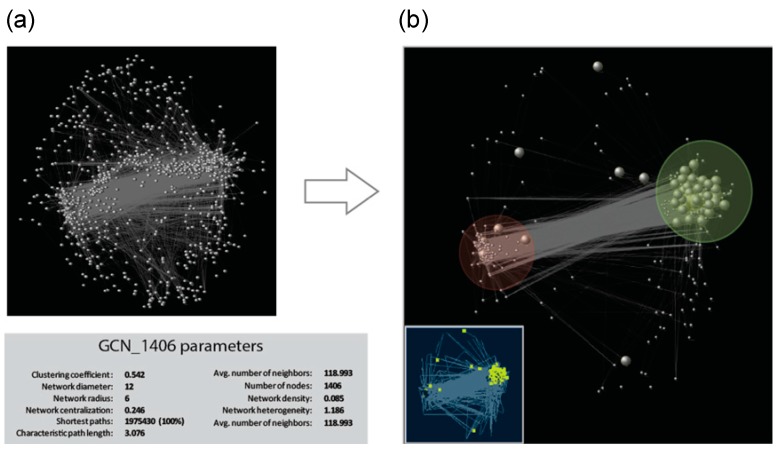
Gene co-expression network identified for the TF-PHOTO genes. (**a**) Global co-expression network and GCN parameters, grey vertices (genes) are connected by an edge if a predefined association between vertices pairs is determined and (**b**) cluster three (in the light green circle) and cluster six (in the light red circle) subnetworks. Bigger grey dots in the big picture, and yellow dots in the smaller picture, represent photosynthesis pathway genes at the core of the TF-PHOTO main global network.

**Figure 5 plants-08-00531-f005:**
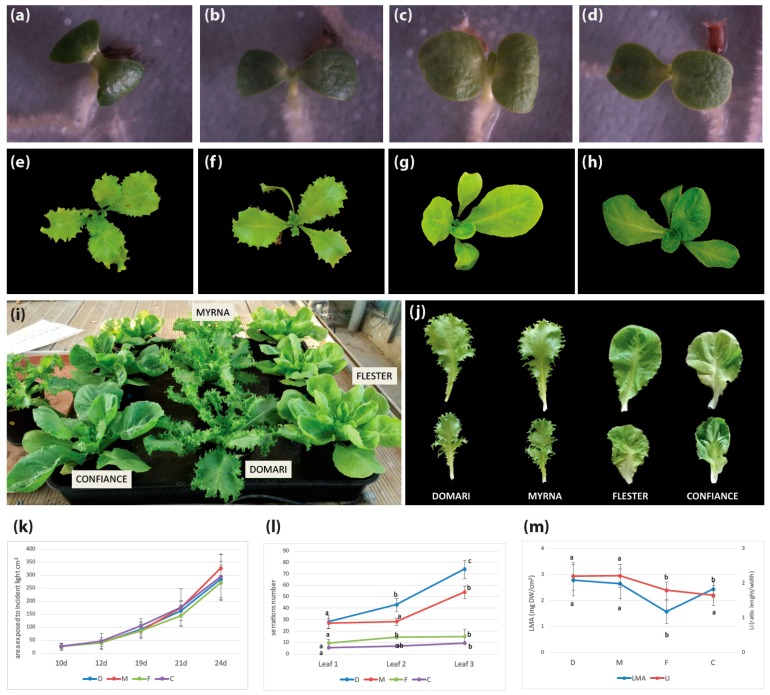
Morphological analysis of *C. endivia* endives, ‘Domari’ (**a**,**e**,**i**,**j**) and ‘Myrna’ (**b**,**f**,**i**,**j**), and escaroles, ‘Flester’ (**c**,**g**,**i**,**j**) and ‘Confiance’ (**d**,**h**,**i**,**j**). (**a**–**d**) Cotyledonary stage, (**e**–**h**) 20-day-old plantlets, (**i**) 40-day-old endive and escarole plants, (**j**) 20th leaf of 50 days mature endive and escarole plants, and (**k**) plant area exposed to incident light in 10- to 24-day-old endive and escarole plantlets (mean ± SD, *n* = 6). Means with different letters represent significant difference at *p* < 0.05. (**l**) Number of serrations on the first, second, and third leaves in endives and escarole 21-day-old plantlets (mean ± SD, *n* = 6). Means with different letters represent significant difference at *p* < 0.05. (**m**) Ratio of leaf dry mass to leaf area (LMA) and ratio of leaf length to leaf width (LI) in fully expanded leaves of 50-day-old endive and escarole plants (mean ± SD, *n* = 6). D, Domari; M, Myrna; F, Flester; C, Confiance.

**Figure 6 plants-08-00531-f006:**
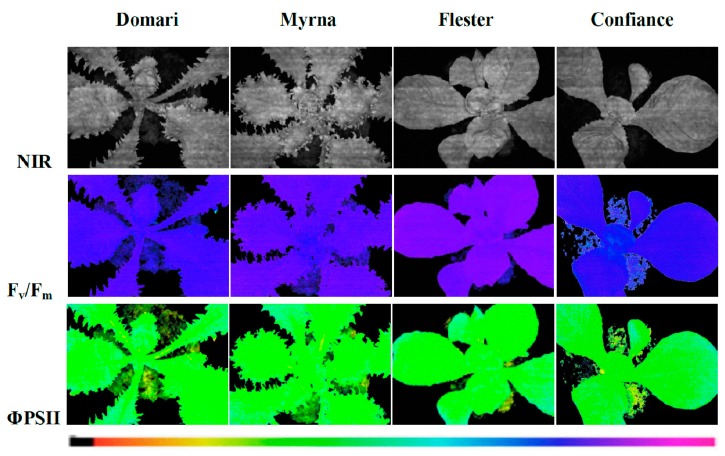
Images of chlorophyll fluorescence parameters in leaves of endives (cv ‘Domari’ and ‘Myrna’) and escaroles (cv ‘Flester’ and ‘Confiance’) cultivars grown under greenhouse conditions. Near-infrared images (NIR) of representative leaves, maximal quantum efficiency (F_v_/F_m_) measured in dark adapted leaves and quantum efficiency of PSII photochemistry (Φ PSII) measured at steady state with light intensity of 370 µmol photons m^−2^ s^−1^ are shown using an Imaging-PAM M-series system. The false colour code depicted at the bottom of the images ranges from 0.000 (black) to 1.000 (pink).

**Figure 7 plants-08-00531-f007:**
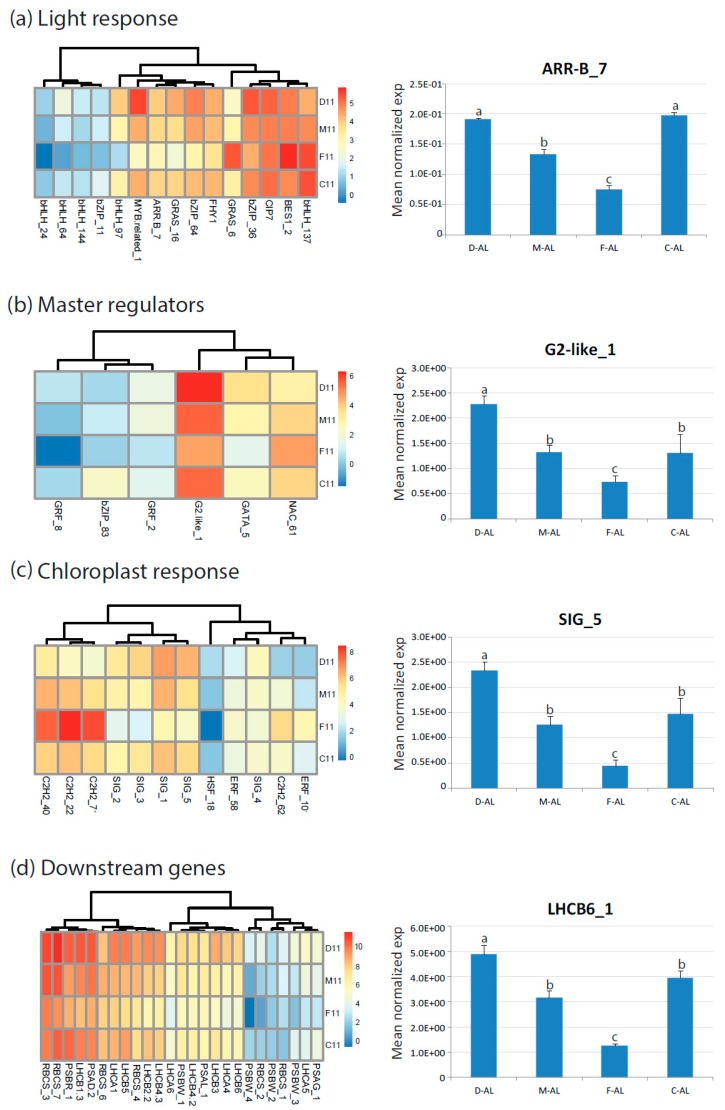
Heat map showing relative expression values (log2 RPKM) of genes differentially expressed in endive and escarole cultivars (year 2011) and related to photosynthesis: (**a**) Light response genes; (**b**) master regulators genes; (**c**) chloroplast response genes; and (**d**) downstream genes (*LHCB, RBC*, photosystems). The expression of one gene for category was reconfirmed by qRT-PCR on RNA from mature leaves (AL). D, ‘Domari’; M, ‘Myrna’; F, ‘Flester’; and C, ‘Confiance’.

**Figure 8 plants-08-00531-f008:**
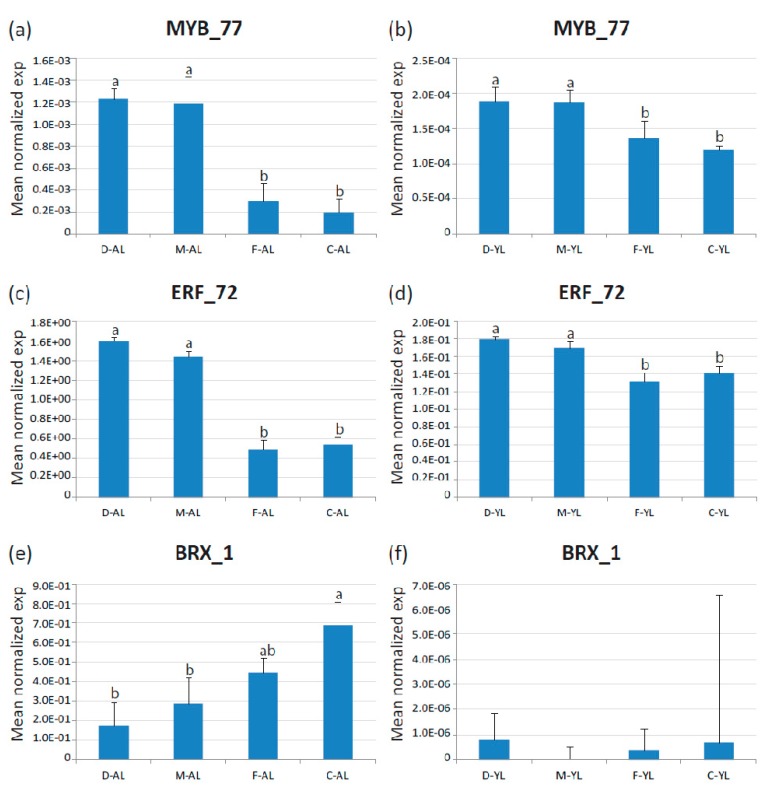
Expression of the three identified TF DEGs in adult (AL) and young (30-day-old, YL) leaves of endives (‘Domari’, D and ‘Myrna’, M) and escaroles (‘Flester’, F and ‘Confiance’, C) analyzed by quantitative real-time PCR. (**a**–**b**) Expression of *CeMYB_77* in adult and young leaves, respectively; (**c**–**d**) expression of *CeERF_72* in adult and young leaves, respectively; and (**e**–**f**) expression of *CeBRX_1* in adult and young leaves, respectively. Data represent the average of two biological replicates with three technical replicates each. Error bars represent SD. Letters indicate significant differences.

**Figure 9 plants-08-00531-f009:**
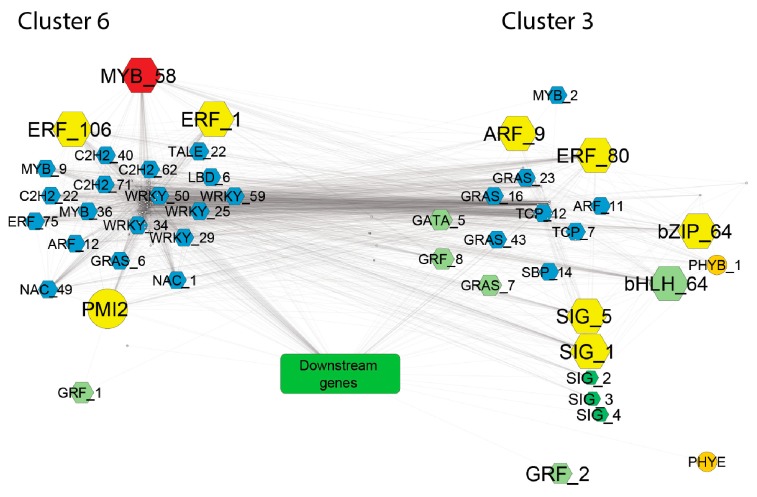
GCN network of photosynthesis and developmental genes from cluster three and six. Downstream genes (represented by the green box) are those involved in the photosynthetic process, which include Rubisco subunits, light harvesting complex, photosynthetic electron transfer, photosynthesis I, and photosystem II. Hexagonal shapes indicate TFs. Colors were used as follows: yellow, top-ranked hub genes; red, the gene with the highest degree in the global TF-PHOTO GCN; light green, master regulators of photosynthesis; dark green, chloroplast response genes; and light blue, developmental genes that may represent novel players in the interplay between leaf development and photosynthesis. Only significant (*p* ≤ 0.05) pairwise correlations with r ≥ |0.9| were considered to obtain this GCN network.

**Table 1 plants-08-00531-t001:** Chlorophyll fluorescence parameters of the four *C. endivia* cultivars.

Cultivar	Parameters^1^					
	F_v_/F_m_ (Relative Units)	Φ PSII (Relative Units)	ETR (μmol elec. m^−2^ s^−1^)	Abs (Relative Units)	Chlorophyll Content (µg cm^−2^)	PRI (Relative Units)
Domari	0.793 ± 0.001 c	0.405 ± 0.004 b	55.73 ± 0.53 b	0.746 ± 0.006 b	19.87 ± 0.96 b	0.043 ± 0.001 a
Myrna	0.807 ± 0.001 b	0.375 ± 0.003 c	49.42 ± 0.47 c	0.715 ± 0.002 c	17.25 ± 1.98 bc	0.038 ± 0.001 ab
Flester	0.828 ± 0.001 a	0.375 ± 0.003 c	46.64 ± 0.47 d	0.675 ± 0.009 d	14.51 ± 1.01 c	0.033 ± 0.001 b
Confiance	0.781 ± 0.001 d	0.419 ± 0.005 a	60.16 ± 0.76 a	0.778 ± 0.002 a	23.57 ± 0.58 a	0.042 ± 0.001 a
*P*	*0.001*	*0.001*	*0.001*	*0.003*	*0.002*	*0.005*

^1^ Chlorophyll fluorescence parameters measured in leaves of endives (cv ‘Domari’ and ‘Myrna’) and escaroles (cv ‘Flester’ and ‘Confiance’) cultivars. F_v_/F_m_, maximal quantum efficiency; ΦPSII, quantum efficiency of PSII photochemistry; ETR, electron transport rate; Abs, photosynthetically active radiation absorptivity; and PRI, photochemical reflectance index (PRI). Different letters in the same column indicate significant differences (*p* ≤ 0.05) according to ANOVA analysis.

**Table 2 plants-08-00531-t002:** List of developmental genes of interest found in cluster three and six that were selected amongst those having a high score to core. Known genes involved in photosynthesis regulations are in green.

Ce ID	Category	Ath ID	Gene Symbol	Score to Core	K	Ce Gene Name
Ce_contig37589	TF	AT4G30080	AUXIN RESPONSE FACTOR 16 (ARF16)	0.925	3	ARF_11
Ce_contig68433	TF	AT2G25180	RESPONSE REGULATOR 12 (RR12)	0.964	3	**ARR-B_7**
Ce_contig18469	TF	AT1G14920	GIBBERELLIC ACID INSENSITIVE (GAI)	0.936	3	**GRAS_16**
Ce_contig31926	TF	AT5G17490	RGA-LIKE PROTEIN 3 (RGL3)	0.972	3	**GRAS_23**
Ce_contig82634	TF	AT4G00150	LOST MERISTEMS 3 (LOM3/SCL6-IV)	0.918	3	GRAS_43
Ce_contig6575	TF	AT4G00150	LOST MERISTEMS 3 (LOM3/SCL6-IV)	0.948	3	GRAS_7
Ce_contig71598	TF	AT3G13960	GROWTH-REGULATING FACTOR 5 (GRF5)	0.941	3	**GRF_8**
Ce_contig83059	TF	AT4G32880	ATHB-8	0.933	3	HD-ZIP_41
Ce_contig21296	TF	AT2G22540	SHORT VEGETATIVE PHASE (SVP)	0.938	3	MIKC-MADS_10
Ce_contig36777	TF	AT5G65060	MADS AFFECTING FLOWERING 3 (MAF3)	0.973	3	MIKC-MADS_13
Ce_contig30145	TF	AT5G60440	AGAMOUS-LIKE 62 (AGL62)	0.936	3	Mtype_MADS_10
Ce_contig1183	TF	AT3G61250	MYB DOMAIN PROTEIN 17 (MYB17)	0.898	3	MYB_2
Ce_contig13510	PHOTO	AT2G18790	PHYTOCHROME B (PHYB)	0.928	3	**PHYB**
Ce_contig73618	TF	AT2G33810	SQUAMOSA PROMOTER BINDING PROTEIN-LIKE 3 (SPL3)	0.899	3	SBP_14
Ce_contig85637	TF	AT2G36990	SIGMA FACTOR 6 (SIG6)	0.955	3	**SIG_5**
Ce_contig49086	TF	AT3G27010	TEOSINTE BRANCHED 1, CYCLOIDEA, PCF (TCP) 20 (TCP20)	0.908	3	TCP_12
Ce_contig33120	TF	AT3G47620	TEOSINTE BRANCHED, CYCLOIDEA, PCF (TCP) 14 (TCP14)	0.986	3	TCP_7
Ce_contig38256	TF	AT1G19850	MONOPTEROS (MP)	0.947	6	ARF_12
Ce_contig18795	TF	AT1G27730	SALT TOLERANCE ZINC FINGER (STZ) (ZAT10)	0.985	6	**C2H2_22**
Ce_contig46000	TF	AT1G27730	SALT TOLERANCE ZINC FINGER (STZ) (ZAT10)	0.955	6	**C2H2_40**
Ce_contig76425	TF	AT1G27730	SALT TOLERANCE ZINC FINGER (STZ) (ZAT10)	0.967	6	**C2H2_62**
Ce_contig81969	TF	AT1G27730	SALT TOLERANCE ZINC FINGER (STZ) (ZAT10)	0.988	6	**C2H2_71**
Ce_contig56990	TF	AT4G34410	REDOX RESPONSIVE TRANSCRIPTION FACTOR 1 (RRTF1)	0.976	6	ERF_75
Ce_contig5939	TF	AT5G48150	PHYTOCHROME A SIGNAL TRANSDUCTION 1 (PAT1)	0.946	6	**GRAS_6**
Ce_contig26571	TF	AT4G37740	GROWTH-REGULATING FACTOR 2 (GRF2)	0.954	6	**GRF_1**
Ce_contig19510	TF	AT5G66870	ASYMMETRIC LEAVES 2-LIKE 1 (ASL1)	0.934	6	LBD_6
Ce_contig49004	TF	AT4G21440	MYB-LIKE 102 (MYB102)	0.974	6	MYB_36
Ce_contig8173	TF	AT2G31180	MYB DOMAIN PROTEIN 14 (MYB14)	0.945	6	MYB_9
Ce_contig7	TF	AT3G10500	NAC DOMAIN CONTAINING PROTEIN 53 (NAC053)	0.971	6	NAC_1
Ce_contig47321	TF	AT5G13180	VND-INTERACTING 2 (VNI2)	0.990	6	NAC_49
Ce_contig85206	TF	AT2G27990	BEL1-LIKE HOMEODOMAIN PROTEIN 8 (BLH8)	0.893	6	TALE_22
Ce_contig23430	TF	AT4G01250	WRKY22	0.957	6	WRKY_25
Ce_contig27558	TF	AT4G23810	WRKY53	0.974	6	WRKY_29
Ce_contig33086	TF	AT3G56400	WRKY70	0.963	6	WRKY_34
Ce_contig60019	TF	AT4G01250	WRKY22	0.909	6	WRKY_50
Ce_contig75830	TF	AT5G13080	WRKY75	0.913	6	WRKY_59

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
