# Peer review of "Transcription Factor Networks in Leaves of Cichorium endivia: New Insights into the Relationship between Photosynthesis and Leaf Development"

_plants, 2019, doi:10.3390/plants8120531_

Round 1
Reviewer 1 Report
I consider that the MS is quite interesting and open new lines of research related to the correlation between photosynthesis and leaf form and development. It is well written and easy to follow.
It will be interesting to include the workflow to generate the gene co-expression network and analysis in a figure to clarify the steps followed.
The MS describes de generation of transcription factor networks from transcriptomic data of Cichorium endivia leaves. The authors analysed the possible relation between leaf development and shape with photosynthesis. The generation of a transcription factor and photosynthesis related genes database (TF-PHOTO) is quite interesting, and the way they proceeded to annotate and cure the database is correct. Using this data, the authors studied the global pattern of gene expression and defined 8 clusters using K-means. Further, they characterize gene co-expression networks (GCN) based in mRNA obtained from two independent field assays. After analysis, they identified 10 hubs genes that belonged to clusters 3 and 6 (as are defined in the MS) including regulators of photosynthesis and hormone response regulators affecting photosynthesis.
The authors performed studies of different photosynthetic parameters and analysed the possible correlation between the narrow curly leaf shape of endives (2 cultivars) and the broad leaves of escaroles (2 cultivars). Although they could not find a clear relationship, the assays are correctly done, and the authors concluded that the observed differences in photosynthetic characteristics are mainly due to cultivar genotypes. Besides, the authors observed that 3 genes are differentially expressed between plants with different shapes.
Finally, the MS describes the GCN of photosynthesis and genes involved in development and conclude that cluster 3 is enriched in positive regulators of photosynthesis while cluster 6 is enriched in genes involved in the response to stress and developmental processes. The methods and approaches used to generate the GCNs are adequate, as well as the rest of the methods described in the MS.
Overall, the MS is well written, and the discussion and conclusions are correct. In my opinion, the main scientific soundness of the MS is the generation of the TF-PHOTO Database that may be very useful and interesting for researchers in the field. Besides, the identification of genes differentially expressed between broad and curly leaves opens, as the authors indicate, new lines of research to study leaf shape. The experimental approaches followed in this study are original, specifically those related to the generation of a transcription factor database and GCNs associating photosynthesis with leaf shape and development.
Minor points:
It will be interesting to include the workflow to generate the gene co-expression network and analysis in a figure. This will help readers to better understand the steps followed to generate the GCNs. A link to the TF-PHOTO data base should be included for readers to explore it.
Author Response
Reviewer 1 Minor points:
It will be interesting to include the workflow to generate the gene co-expression network and analysis in a figure. This will help readers to better understand the steps followed to generate the GCNs. A link to the TF-PHOTO data base should be included for readers to explore it.
Response to Reviewer 1:
We would like to thank the Reviewer for his interest in the manuscript and nice comments. We included a new Figure (Figure 3) that illustrates the workflow to generate the gene co-expression networks and analyses. There is not a web site to access TF-PHOTO, rather all the information about the database, including nucleotide and amino acid sequences, categories, annotations, Arabidopsis homologues IDs, cluster assignment, ranked genes and RPKM original values, has been provided and is available in Table S1. We then asked the publishers to insert a hyperlink in the legend of Figure 3 to directly access Table S1. We also reloaded Table S1 as the category column format was not homogeneous, and Table S3 that now provides the two flatten matrix of the pairwise correlation data for both r ≥ |0.8| (used for the global GCN) and r ≥ |0.9| (used for the subnetwork from which Cluster 3 and Cluster 6 genes were extracted to highlight the relationship between developmental genes and photosynthesis in Figure 8, now Figure 9).
Reviewer 2 Report
The paper entitled “Transcription factor networks in leaves of Cichorium endivia: new insights in to the relationship between photosynthesis and leaf development” describes the investigation of possible links between photosynthesis and leaf shape variation by combination of transcriptome, physiological and molecular analyses. By using excellent model plant, C. endivai, authors succeeded in revealing the transcription factors which might have important roles in regulating photosynthesis and leaf morphology.I think that this paper is interesting and important, because the results would provide new insights into the relationship between leaf shape and function. I believe that I believe that this paper would attract wide range of readers. As a conclusion, the paper is sufficient to merit publication in Plants.
The experiments and analyses are carefully performed and the conclusions are reasonable. I do not see big problem for this paper and I think it is almost ready for publication except for the following minor comments.
Title
Cichorium Endivia -> Cichorium endivia
Figures
In most of the figures, the captions are too small to see.
Author Response
Reviewer 2 minor comments:
Title
Cichorium Endivia -> Cichorium endivia
Figures
In most of the figures, the captions are too small to see.
Response to Reviewer 2:
We would like to thank you for the nice comments and interest in our paper. We changed the text according to your comments and increased the captions up to the limit allowed by previous formats and layouts. We also increased figure quality so that all the text can be read, either at 100% scale or at higher zoom level without losing resolution.